# Comparison of Growth Curve Estimates of Infants in São Tomé Island, Africa, with the WHO Growth Standards: A Birth Cohort Study

**DOI:** 10.3390/ijerph16101693

**Published:** 2019-05-14

**Authors:** Marisol Garzón, Ana Luísa Papoila, Marta Alves, Luís Pereira-da-Silva

**Affiliations:** 1Tropical Clinic Teaching and Research Unit, Instituto de Higiene e Medicina Tropical, Universidade NOVA de Lisboa; Lisbon 1349-008, Portugal; garzon.marisol1@gmail.com; 2Global Health and Tropical Medicine R&D Center, Instituto de Higiene e Medicina Tropical, Universidade NOVA de Lisboa, Lisbon 1349-008, Portugal; 3Research Unit, Centro Hospitalar Universitário de Lisboa Central, Lisbon 1169-045, Portugal; ana.papoila@nms.unl.pt (A.L.P.); marta.alves@chlc.min-saude.pt (M.A.); 4Centre of Statistics and its Applications, University of Lisbon, Lisbon 1749-016, Portugal; 5Medicine of Woman, Childhood and Adolescence Teaching and Research Area, NOVA Medical School, Universidade NOVA de Lisboa, Lisbon 1169-056, Portugal

**Keywords:** birth cohort, breastfeeding, growth charts, infant growth, lower-middle-income country, Republic of São Tomé and Príncipe, WHO standards

## Abstract

This birth cohort study compared the infant growth curve estimates in São Tomé Island to the WHO growth standards. Despite this island belonging to a lower-middle-income country, there were several factors favorable for growth that were present. Four-hundred and seventy-five full-term singleton appropriate for-gestational-age infants were enrolled and followed-up to 24 months of age. Weight-for-age, length-for-age, weight-for-length, body mass index-for-age, head circumference-for-age, weight velocity, and length velocity curves were estimated and compared to the WHO standards. In the first 6 months of age, the weight gain was adequate in the presence of a high prevalence of exclusive breastfeeding. Thereafter, weight trajectories tracked close to the WHO standards, except for a progressive decline in the infants growing in higher percentiles, especially in girls. Median length at birth was below the median WHO standards, followed by an early postnatal velocity spurt, which probably reflected the transition from an unfavorable to a more favorable postnatal environment. Thereafter, linear growth faltering was observed without relevant deterioration up to 24 months of age, which was probably due to the presence of protective factors. These results may be useful to implement strategies to further approximate infant growth in São Tomé Island to the WHO standards.

## 1. Introduction

### 1.1. Growth in Lower Middle Income Countries

The main identified determinants that affect growth in lower-middle-income countries (LMICs) include maternal and fetal undernutrition, low prevalence of breastfeeding, adverse social and economic factors, and morbidity due to infectious diseases [1,2,3].

A recent birth cohort study published by our team [4] assessed the association between enteric parasitic infections and growth faltering in the infants in São Tomé Island, which belongs to the Republic of São Tomé and Príncipe, an African lower-middle-income country. We found that the great majority of the infants were asymptomatic. The multivariable analysis identified statistically significant associations between some parasitic infections and linear growth although these associations were considered to only have a mild magnitude [4]. Moreover, as approximately one-fifth of households were classified as being deprived, the association between deprivation and growth was studied and confirmed. For this purpose, a multidimensional poverty index that includes education, health, and standard of living dimensions [5] was used. Compared to other LMICs, we identified that this cohort of infants was under conditions that were favorable for growth, similar to those used by the World Health Organization (WHO) Multicenter Growth Reference Study to construct the standard growth charts [1,3]. These included a high prevalence of exclusive breastfeeding until 6 months of age, introduction of complementary foods by 6 months of age, continued breastfeeding to 12 months of age, and absence of significant morbidity [4]. Furthermore, the studied infants benefited from deworming programs and feeding advice while a pediatrician attended any acute events in these infants. These factors that are favorable for health and growth may explain the mild magnitude of growth faltering associated with parasitic infections, which were found to be mostly asymptomatic [4].

### 1.2. WHO Standards

The WHO Multicenter Growth Reference Study has developed growth standards that describe the growth of healthy breastfed infants living in six countries from diverse geographical regions with socioeconomic conditions that are favorable for growth, which assumes that breastfeeding is the biological norm for growth and development [6,7,8]. The growth patterns from populations with widely different nutritional status profiles have been compared both to the WHO standards and the National Center for Health Statistics/WHO international growth reference and it was concluded that the WHO standards provide a better tool for monitoring the rapid and changing rate of growth in early infancy [9]. The WHO standards demonstrate for the first time ever that when given an optimum start in life, children born in different regions of the world have the potential to grow to reach the same ranges of weight and height for their age [6]. Hence, the WHO standards should be a norm to monitor the growth of every child worldwide, regardless of ethnicity, socioeconomic status, and type of feeding. Furthermore, they are being adopted by an increasing number of countries and their use has been endorsed by the main scientific bodies [10]. 

Nevertheless, as child growth may be suboptimal in LMICs that have been identified to experience multifactorial growth faltering, the use of WHO growth standards may not be straightforward in these settings.

### 1.3. Objective

The aforementioned described scenario motivated the present study to compare birth cohort growth curve estimates in São Tomé Island with the WHO growth standards [7] in the presence of important protective factors that are usually absent or insufficient in LMICs.

## 2. Materials and Methods

### 2.1. Study Design and Setting

This study is a secondary analysis nested within a broader birth cohort study conducted in São Tomé Island (belonging to a sub-Saharan archipelago) from March 2013 to July 2015. The protocol specificities of this broader study have been previously described [4,11]. 

Consecutive singleton full-term (≥37 weeks of gestation) appropriate-for-gestational-age neonates (>10th and <90th percentiles) [12] were eligible for our birth cohort, following the criteria that were used for the construction of WHO growth standard curves [7]. From the initial convenience sample of 500 neonates, 25 were excluded due to preterm birth, low birth weight, lack of gestational age information, major congenital malformations, and perinatal asphyxia. Thus, only 475 (95%) were enrolled in the birth cohort, corresponding to approximately 8.6% of live births in São Tomé based on a contemporary census report (Instituto Nacional de Estatística—INE STP 2012). All recruited infants were native Africans, from three districts where 50.5% of the population of São Tomé lives. Although named a birth cohort, this study recruited infants during the neonatal period at a median (interquartile range) age of 0.30 (0.20–0.50) months [4]. 

Study data were obtained from a baseline visit performed during the neonatal period and thereafter from visits scheduled monthly during the first year of age and bimonthly during the second year. The visits took place at the main health care centers at the outpatient facilities, and at local hospitals in the three districts involved [4].

### 2.2. Birth Cohort and Anthropometry

Varying proportions of the infants missed the scheduled visits (Figure 1), with some of the infants not present at a certain point of assessment but returning for the next assessment.

The anthropometry assessment was performed by the same trained observer (MG) according to the techniques recommended by the WHO [6]. The infants were weighed using an electronic infant scale (Seca 334, GmbH & Co. KG, Hamburg, Germany) to the nearest decigram, and the crown–heel length was measured using an infantometer (Seca 207, GmbH & Co. KG, Hamburg, Germany) to the nearest millimeter. Baseline measurements were taken during the neonatal period and thereafter monthly during the first year of age and bimonthly during the second year. For the curve analysis of weight-for-age, length-for-age, weight-for-length, body mass index (BMI) for-age, and head circumference-for-age, the infants with a single observation were excluded which resulted in a final sample size of 414 infants. Regarding weight and length velocities, only infants with more than one bimonthly increment were considered, which resulted in a final sample size of 372 infants. 

### 2.3. Statistical Methods

The sex-specific growth curves based on 5338 longitudinal observations, from the neonatal period to 24 months of age were compared to the WHO growth standards [7,9]. The anthropometric measurements included length-for-age, weight-for-age, weight-for-length, BMI-for-age, and head circumference-for-age. The statistical methods for constructing the growth curves were based on the WHO approach [6]. To identify the starting distributions and corresponding degrees of freedom for the smoothers, the *lms* function from GAMLSS package (Generalized Additive Models for Location, Scale, and Shape) [13] was used for each anthropometric measurement (weight-for-age, length-for-age, weight-for-length, BMI-for-age, and head circumference-for-age). This function estimates the power λ of the transformation of age $x^{\lambda }$ and chooses between the distributions of Box-Cox normal [14], Box-Cox power exponential [15], and Box-Cox t [16]. The parameters of these distributions include the median µ (location), the coefficient of variation σ (scale), the skewness ν (Box-Cox transformation power), and the kurtosis τ (parameter related to kurtosis), with the exception of Box-Cox normal distribution that is not characterized by the parameter τ.

According to the WHO proposal [6], the GAMLSS models were then fitted to each anthropometric measurement considering the previously obtained starting distributions and λ, with cubic splines used as smoothers. These simple starting models, which consider the modeling of µ and σ with $x^{\lambda }$ as the only covariate, were implemented to finetune the degree of smoothness corresponding to the age transformation. For this purpose, possible combinations were considered and the criteria used to choose among these models were part of the AIC (Akaike Information Criterion) and its generalized version that has a penalty equal to 3 (GAIC(3)) [15]. 

Considering these initial results, the new GAMLSS models were fitted to the data and their goodness-of-fit was assessed with residual plots, worm plots, and Q-statistics [13]. Regarding the worm plots (to determine if the model fits adequately across all ages), the model has a good overall fit if no specific shape is depicted by the points and all observations fall inside the two elliptic curves. Q-statistics are another approach to test the normality of the residuals for each age range and indicate if any inadequacy exists regarding the models for the parameters µ, σ, ν, and τ. They provide information about the magnitude of the residuals and significant *p*-values mean the significant inadequacy of the model.

If Q-statistics indicate an inadequacy regarding the fit of ν or τ distribution parameters (with significant *p*-values), the new GAMLSS models considering the modeling of these parameters were fitted to the data and the corresponding smoothness degrees of freedom were identified (keeping µ and σ curves degrees of freedom). Still regarding the goodness-of-fit of the final models, the calibration plots that compare the smoothed estimates with the empirical centiles were constructed.

Finally, the last model was obtained according to the WHO approach [6], after confirming the appropriateness of the distribution function (Box-Cox t, Box-Cox Cole and Green, Normal, Johnson’s SU, Generalized Gamma, and Box-Cox power exponential distributions were compared using GAIC(3)). A new search for the final age transformation power λ was also performed using a grid of values ranging from 0.05 to 1 in intervals of 0.05. The λ that performed better was chosen based on the global deviance criterion. 

For comparing the growth curve estimates to the WHO growth standards [7,9], we estimated the 3rd, 25th, 50th, 75th, and 97th percentile smooth curves.

The weight and length velocities were also studied using bimonthly increments. However, due to the instability of the modeling process regarding these variables, the age transformation power λ and the degrees of smoothing for µ, σ, and ν, were the same as those used by WHO [17]. 

The optimal models that provided the best fit to generate the growth curves are provided as follows.

For length-for-age:BCPE (λ = 0.35, df(µ) = 7, df(σ) = 2, ν = 1, τ = 2) for boysBCPE (λ = 0.35, df(µ) = 7, df(σ) = 2, ν = 1, τ = 2) for girls

For weight-for-age:BCPE (λ = 0.35, df(µ) = 8, df(σ) = 2, df(ν) = 2) for boysBCPE (λ = 0.10, df(µ) = 8, df(σ) = 2) for girls

For weight-for-length:BCPE (df(µ) = 6, df(σ) = 1, df(ν) = 1, τ = 2) for boysBCPE (df(µ) = 5, df(σ) = 3, df(ν) = 1, τ = 2) for girls

For BMI-for-age:BCPE (λ = 0.30, df(µ) = 7, df(σ) = 2) for boysBCPE (λ = 0.31, df(µ) = 8, df(σ) = 2) for girls

For head circumference-for-age:BCPE (λ = 0.64, df(µ) = 5, df(σ) = 2, df(τ) = 1) for boysBCPE (λ = 0.09, df(µ) = 7, df(σ) = 3, ν = 1, τ = 2) for girls

For 2-month interval weight velocity:BCPE (λ = 0.05, df(µ) = 12, df(σ) = 6, df(ν) = 3, τ = 2) for boysBCPE (λ = 0.05, df(µ) = 12, df(σ) = 5, df(ν) = 4, τ = 2) for girls

For 2-month interval length velocity:BCPE (λ = 0.05, df(µ) = 9, df(σ) = 7, df(ν) = 1, τ = 2) for boysBCPE (λ = 0.05, df(µ) = 10, df(σ) = 7, df(ν) = 1, τ = 2) for girls

Using these final GAMLSS models, we constructed several tables with percentiles (ranging from the 1st to the 99th percentiles) and estimates of LMS parameters (data not shown). 

As an example, the goodness-of-fit study length-for-age model for boys, which was based on residual analysis, worm plots, Q-statistics, and calibration plots, is presented in the Appendix A.

A level of significance α = 0.05 was considered. Data were analyzed using R (R: A Language and Environment for Statistical Computing, R Core Team, 2019; R Foundation for Statistical Computing, Vienna, Austria, http://www.R-project.org).

## 3. Results

### 3.1. Factors Affecting Growth

Compared with infants who prematurely dropped-out, those completing the study had significantly higher weight and length at birth, belonged predominantly to a district with a higher wealth index, and their mothers were older and received longer education [4].

#### 3.1.1. Socioeconomic Status and Feeding Practices

Regarding the socioeconomic status, 287 (60.8%) out of the 475 households approached provided data for the multidimensional poverty index scoring [5]. In the education dimension, 46.3% of mothers attended more than five years of school. In the health dimension, child mortality occurred in 2.8% of households; and in the living standard dimension, 99.7% of households had improved water sources, 99.3% had a finished floor, 88.9% cooked with solid fuel, 86.1% had electricity, 75.6% had assets, and 66.2% had improved sanitation. The total multidimensional poverty index score rated 24.0% of households as being deprived, and of these deprived households, one-third were determined to be severely deprived [4]. 

Practically all infants (99.8%) were ever breastfed during the study period; most of them (88.4%) were exclusively breastfed during the first 6 months; 64.8% and 13.9% were breastfed up to 1 year and 2 years of age, respectively. Complementary foods were introduced at a mean age of 6 months, and in 21.2% of infants they were introduced before [4]. 

#### 3.1.2. Morbidity

The morbidity recorded at the points of assessment was relatively low during the study period [4]. Episodes of acute diarrhea were recorded in less than 5.2% of the infants during the first 5 months of age and 6.8–15.1% after 5 months of age. Acute respiratory infections (mostly upper respiratory infections) were recorded in less than 3.2% during the first 2 months of age and 19.9–32.8% after 2 months of age. Malaria was diagnosed in four (0.8%) infants only after 12 months of age while the HIV test was positive in one infant. The recorded chronic conditions included allergic symptoms in 174 infants and anemia in 133 (6 with sickle cell anemia). During the study period, three deaths occurred with clinical diagnoses (without autopsy) of heart failure at one month of age; acute hemorrhagic syndrome due to poisoning at 13 months of age; and pneumonia at 16 months of age. In addition to the satisfactory breastfeeding rates, adequate age for the introduction of complementary foods and relatively low morbidity, other important factors may have improved the nutritional status and growth of the studied infants. The principal observer (MG, an experienced pediatrician) provided feeding advice and attendance for acute events. The infants older than one year received mebendazole every four months in compliance with the WHO preventive chemotherapy strategy for soil-transmitted helminths [18] implemented in São Tomé and Principe. According to the current recommendations [19], the infants were also treated for *Giardia lamblia* with metronidazole in the case of the microscopic detection of trophozoites (regardless of symptoms) or a positive rapid test. 

### 3.2. Abthropometry

The median, minimum, and maximum for weight-for-age, length-for-age, weight-for-length, BMI-for-age, and head circumference-for-age measured at each point of assessment are included in Appendix A.

Differences between WHO percentiles and smoothed centile curves at 24 months of age for weight-for-age, length-for-age, weight-for-length, BMI-for-age, and head circumference-for-age are presented in Table 1.

The percentages of infants above the 90th percentile and below the 10th percentile of the WHO standards for weight-for-age, length-for-age, BMI-for-age, and head circumference-for-age, at the neonatal period, 6 months, 12 months, and 24 months of age are presented in Table 2.

#### 3.2.1. Weight-for-Age 

In both sexes, the weight-for-age curves tracked close to the WHO standards [7] from the neonatal period to approximately 6 months of age (Figure 2). Beyond this age, these curves showed a progressive downward deviation from the WHO standards [7] (Figure 2), particularly for the infants growing in higher percentiles (97th and 75th percentiles) and were more pronounced in girls (Figure 2A). In fact, for the 97th percentile, the differences reached a maximum of 1.2 kg for boys and 1.4 kg for girls at 24 months of age (Table 1).

#### 3.2.2. Length-for-Age 

In both sexes, the length-for-age curves tracked below to the WHO standards [7] from the neonatal period to 24 months of age (Figure 3). At this age, no differences between sexes were identified for any percentiles and the differences with WHO standards [7] were constantly around 1.8 cm for all percentiles (Table 1).

#### 3.2.3. Weight-for-Length 

In both sexes, the weight-for-length curves tracked close to the WHO standards [7], except for a progressive downward deviation in the infants measuring more than around 85 cm and growing in higher percentiles (97th and 75th percentiles) (Figure 4). For the 97th percentile, the deviation was more pronounced in girls (Figure 4B) and reached a maximum difference of 0.8 kg at 95.5 cm compared to 0.3 kg at the same length in boys (Table 1).

#### 3.2.4. Body Mass Index-for-Age

In both sexes, the BMI-for-age curves tracked quite close to the WHO standards [7] (Figure 5). Of note, the curves tracked slightly above the WHO standards [7] from the neonatal period to 6 months of age. Beyond this age, a crossover for all percentiles was observed (Figure 5). At 24 months of age, a downward deviation in relation to the WHO standards [7] was noticed for higher percentiles, reaching a difference of 0.6 kg/cm^2^ for both sexes (Table 1).

#### 3.2.5. Head Circumference-for-Age 

In both sexes, the head circumference-for-age curves tracked close to the WHO standards [7] (Figure 6). However, for the infants in the 3rd percentile, the trajectories of the curve tracked slightly above and parallel to the WHO standards [7]. At 11 months of age, a downward crossover was observed for the 50th, 75th, and 97th percentiles for boys and the 97th percentile for girls. At 24 months of age, slight deviations from the WHO standards [7] were noticed for lower percentiles, which reached a difference of 0.5 cm for the 3rd percentile in both sexes (Table 1).

#### 3.2.6. Two-Month Weight Velocity

In both sexes, an increase in the weight velocity was observed from the neonatal period to around three months of age. This early increase is not observed in the WHO standards [19] (Figure 7). From this age up to 15 months, the velocity curve tracked below the WHO standards in both sexes [19], especially for the 3rd percentile. From 15 to 24 months of age, the velocity curves tracked close to the WHO standards in boys, and the velocity curves fluctuated slightly around the WHO standards in girls [19].

#### 3.2.7. Two-Month Length Velocity

In both sexes, an increase in the length velocity was observed from the neonatal period to around four months of age. This early increase is not observed in the WHO standards [19] (Figure 8). From this age up to 15 months, the velocity curve tracked close to the WHO standards [19] at the 97th percentile. For the remaining percentiles, a downward deviation was noticed. A lower percentile resulted in a more pronounced difference, especially for the 3rd percentile. From 15 to 24 months of age, the velocity curves tracked close to the WHO standards [19].

## 4. Discussion

### 4.1. Cohort Study Specificities

This is a nested study in the first birth cohort study ever to be performed in São Tomé [4]. Only singleton term appropriate-for-gestational age neonates were included, which corresponded to 8.6% of live-births in São Tomé. All enrolled infants were native Africans. Regarding the socioeconomic status, 24.0% of the households were deprived and from these deprived households, one-third were found to be severely deprived [4]. Particularly, 37.8% of the households had unimproved sanitation, which is recognized as a leading risk factor for growth stunting in developing countries [3]. Despite these adverse factors, we found conditions that were favorable for growth, as described by the WHO Multicenter Growth Reference Study [6], including a high prevalence of exclusive breastfeeding at 6 months of age, continued breastfeeding up to 12 months of age, the introduction of complementary foods by 6 months of age, and the absence of significant morbidity, including acute infectious diseases [4]. Particularly, the vast majority of the infants with diagnosed enteric parasitic infections were asymptomatic, which might be related to anti-parasite programs adopted in São Tomé and Principe that included preventive chemotherapy for soil transmitted helminths and metronidazole for detected cases of *Giardia lamblia* [4,11]. In addition, during the scheduled points of assessment, the infants benefited from feeding advice and attendance for acute events provided by an experienced pediatrician [4]. Finally, it can be speculated that some health benefits might have resulted from the “Hawthorne effect” due to the mothers’ behavior being modified in response to their awareness of being observed [20]. 

In the aforementioned study by Garzón et al. [4], the anthropometry of each infant was compared to the WHO standards [7] using z-scores. In the present study, the longitudinal cohort growth was compared to the WHO standards [7] with the purpose of evaluating infant growth in the presence of conditions that were favorable for growth [6].

### 4.2. Body Weight Curves 

A remarkable finding in the present study was a pattern of satisfactory weight gain in the first 6 months of age, which was indicated by the weight-for-age and weight-for-length curves that tracked close to the WHO standards [7]. Nevertheless, weight velocity tracked below the WHO standards for most of the percentiles.

The high rate (88.4%) of exclusive breastfeeding at 6 months of age [4] was substantially higher in this cohort compared to the finding of 37% from other surveys in LMICs [21]. These findings confirm the high protective effect of breastfeeding in low-income settings during the early vulnerable period of life [21,22]. In this study, the infants’ nutritional status was better described using the BMI-for-age curves. During the first 6 months of age, the trajectories of BMI-for-age curves tracked slightly above the WHO standards [7], which probably revealed the genetic growth potential of this population when they benefited from exclusive breastfeeding. Exclusive breastfeeding is considered to be the gold standard nutrition for the first 6 months of age [21]. After this age, the trajectories of the weight-for-age, weight-for-length, and BMI curves tracked slightly below the WHO standards, except for a marked decline in the infants growing in higher percentiles. Although the majority (64.8%) of the studied infants were still breastfed at 12 months of age [4], this decline may reflect a loss of the full growth potential after the introduction of complementary feeding. As the reported morbidity, including acute infectious events, was low in this cohort [4], the aforementioned suboptimal weight gain, particularly reflected by low weight velocity, may be attributable to the possible poor quality of complementary foods (not assessed) and/or to suboptimal socioeconomic status, considering that poor wealth status is associated with decreased attained growth in children from LMICs [1]. The more pronounced downward deviations of the weight-for-age and weight-for-length curves of girls growing in higher percentiles might be explained by the relative neglect of girls in the intra-household allocation of nutrients (not assessed) reported in relation to boys [23].

### 4.3. Linear Growth 

A pattern of attained linear growth faltering from the neonatal period to 24 months of age was observed, which was characterized by the length-for-age curves tracking below the WHO standards [7]. In this cohort, some deficit in length has been previously reported in the neonatal period, with mean length z-scores of −0.73 and −0.68 for girls and boys, respectively, despite mean weight z-scores of 0.03 and −0.04 for girls and boys, respectively [4]. A neonatal length deficit has also been described in the poorer regions of Southeast Asia and Africa, with length-for-age z-scores of −0.75 in Asia and −0.35 in Africa. This further deteriorated and reached a z-score of −1.5 at 24 months of age [2]. This differs from our cohort in which a mean length-for-age z-score of -0.71 with few oscillations was reported up to 24 months of age [4].

As the attained growth only provides a cumulative measure of growth, the length velocity was used to refine the analysis of the linear growth pattern [19,24]. An initial velocity spurt was observed from the neonatal period to around four months of age, in both sexes; this spurt was accompanied by a similar weight velocity trend. From this age, the length velocity decelerated and the length velocity curves tracked below the WHO standards until 15 months. From 15 to 24 months of age, the velocity trajectories tracked close to the WHO standards [7]. 

We speculate that the transition from an intrauterine environment restraining linear growth to a more favorable postnatal environment supported by optimal nutrition (exclusive breastfeeding) at this age probably explains the early postnatal length velocity spurt. Beyond 15 months of age, the conditions that are favorable for growth may have enabled the achievement of the genetic growth potential reflected by a normal length velocity, which is in contrast to the progressive linear growth deterioration described in other LMICs [2]. In this context, it is difficult to know to what extent the linear growth pattern in São Tomé reflects the local environmental factors or the genetic background [25]. In fact, as the attained length reflects the health stock accumulated through social and environmental exposures, maternal stature has been considered to be a stable and useful marker for assessing intergenerational linkages in health [25,26]. In our cohort, a relatively high proportion (41.0%) of mothers measured more than 160 cm and only 0.6% measured less than 145 cm. Interestingly, African women have been reported to be taller than Asian and Latin American women despite African women having a low income [27]. Furthermore, in our cohort, an adjusted positive association between the height of the mothers and the linear growth of the offspring was reported [9]. This corroborates the positive association between the maternal height and offspring growth beyond the neonatal period, which was previously reported by other authors [26,28]. 

### 4.4. Head Growth

In both sexes, the head circumference-for-age curves tracked slightly above the WHO standards [7] with a further crossover in some percentiles, although the deviations were clinically irrelevant. As only appropriate for-gestational-age infants who were born at full term were recruited and they had not severe postnatal undernutrition [4], it was expected that the nutritional status would not affect head growth. Therefore, we speculate that the estimated head circumference-for-age curve trajectories are related to the genetic background. 

### 4.5. Limitations

Although a relatively large sample size was attained for this national cohort study, it only represents 8.6% of the live births in São Tome and may not be considered to be representative of the whole population. Moreover, varying proportions of the infants missed the scheduled visits, with many of the infants not present at a certain point of assessment but returning for the next assessment (Figure 1). Despite this limitation, the goodness-of-fit of the resulting models was highly satisfactory. Since birth, anthropometry was not available in our study, the neonatal anthropometric measurements undertaken at recruitment at a median age of 0.30 months were compared to the WHO birth data standards [7]. A selection bias should be acknowledged. Infants completing the study had significantly higher weight and length at birth, a higher wealth index, and their mothers were older and received longer education than infants who prematurely dropped out. On the other hand, this bias has further contributed to approximate the subgroup of infants completing the study to the criteria that have been used for the construction of WHO growth standards curves [6].

### 4.6. Strengths

This was the first birth cohort study ever to be performed in the Republic of São Tomé and Príncipe. To the best of our knowledge, this study is a pioneer in analyzing infant growth in a LMIC, which benefits from several important conditions that are favorable for growth. These conditions were determined by the WHO Multicenter Growth Reference Study [6] and are usually absent in other LMICs. The feeding advice and attendance for acute events provided by a pediatrician may have contributed to attaining these favorable conditions. Moreover, the aforementioned selection bias further approximated the subgroup of infants completing the study to the criteria used for the construction of WHO growth standards curves [6].

For data modeling, the same methodology used by the WHO [6] was adopted in our cohort study, which allowed the obtained results to be compared to the WHO standards [7].

## 5. Conclusions

This birth cohort study suggests that, despite living in suboptimal socioeconomic conditions, the infants in São Tomé Island have satisfactory growth, as reflected by the estimates of the weight curves, especially those of the BMI, which is likely due to the presence of several conditions that are favorable for growth. This is especially noticeable within the first 6 months of age when exclusive breastfeeding had a major influence.

During the study period, a pattern of attained linear growth faltering was observed although the deterioration was quite less than described in the majority of LMICs. It is noteworthy that low neonatal length z-scores were recorded, which was followed by a length velocity spurt from the neonatal period up to around four months of age. This may be explained by the transition from a restrictive intrauterine environment to a more favorable postnatal environment supported by exclusive breastfeeding. 

In the absence of prenatal or postnatal severe undernutrition, the head growth was not affected as expected and the slight deviations from the standards may be related to the genetic background. 

This study provides useful data for stakeholders and health professionals to guide future health interventions that aim to further approximate infant growth in São Tome Island according to the WHO standards.

## Figures and Tables

**Figure 1 ijerph-16-01693-f001:**
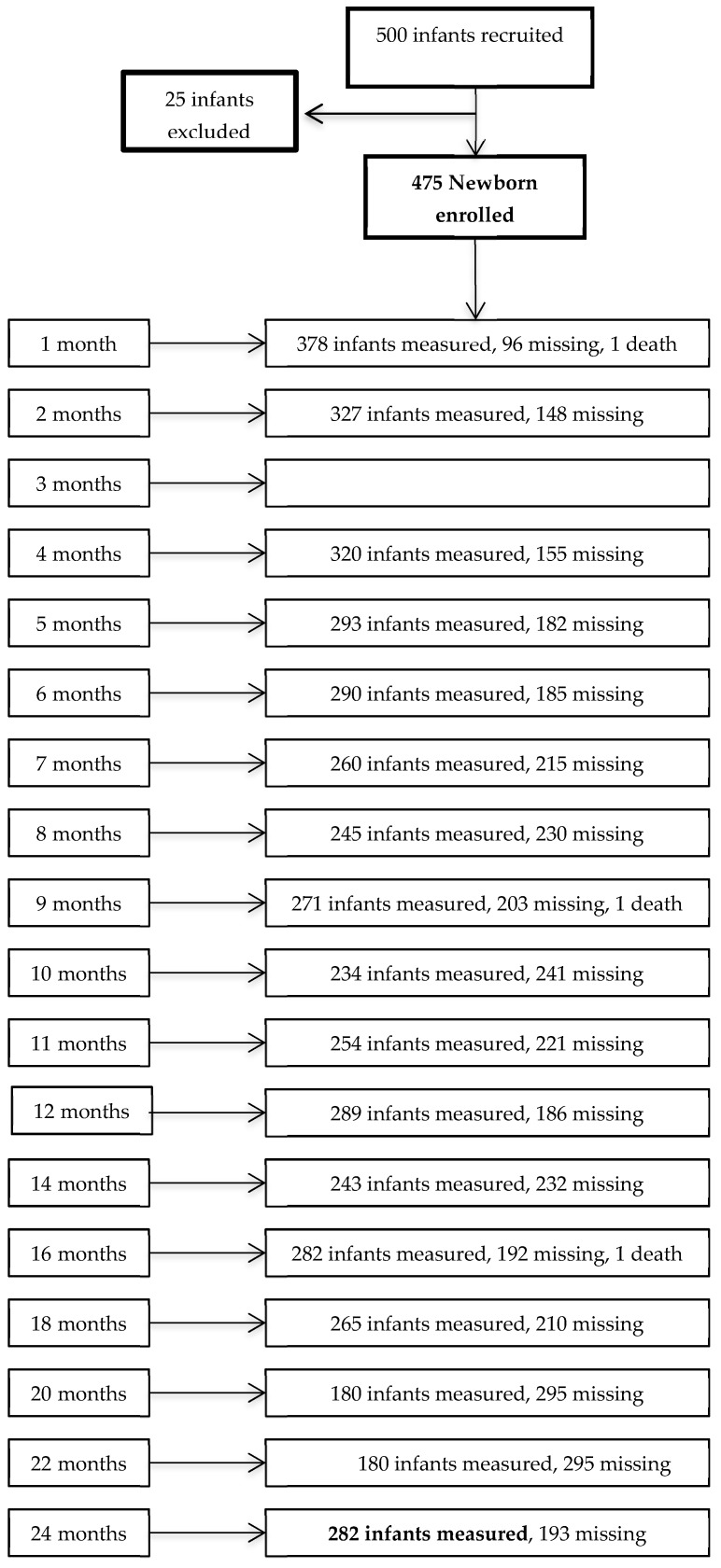
Flow-chart showing the number of infants measured and missing at each point of assessment during the study period.

**Figure 2 ijerph-16-01693-f002:**
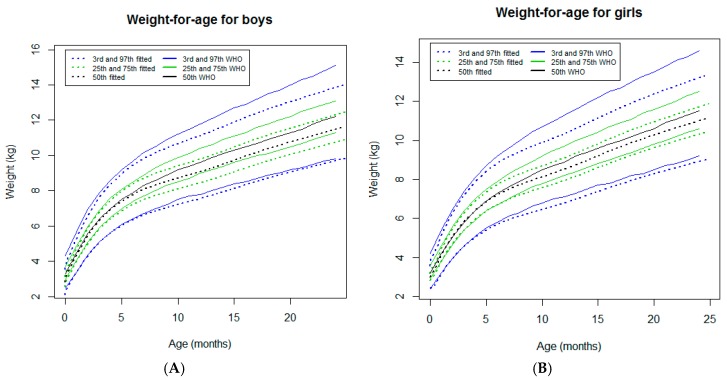
Comparison of the WHO Child Growth Standards (solid lines) with smoothed centile curves (dotted lines) of measured weight-for-age for boys (**A**) and girls (**B**) from the neonatal period to 24 months of age.

**Figure 3 ijerph-16-01693-f003:**
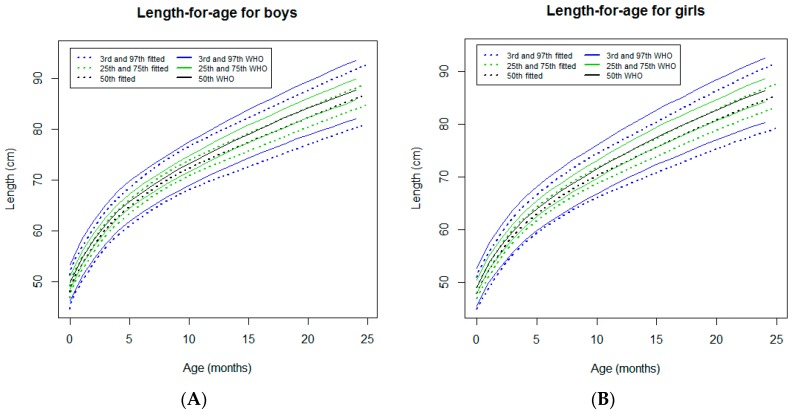
Comparison of the WHO Child Growth Standards (solid lines) with smoothed centile curves (dotted lines) of measured length-for-age for boys (**A**) and girls (**B**) from the neonatal period to 24 months of age.

**Figure 4 ijerph-16-01693-f004:**
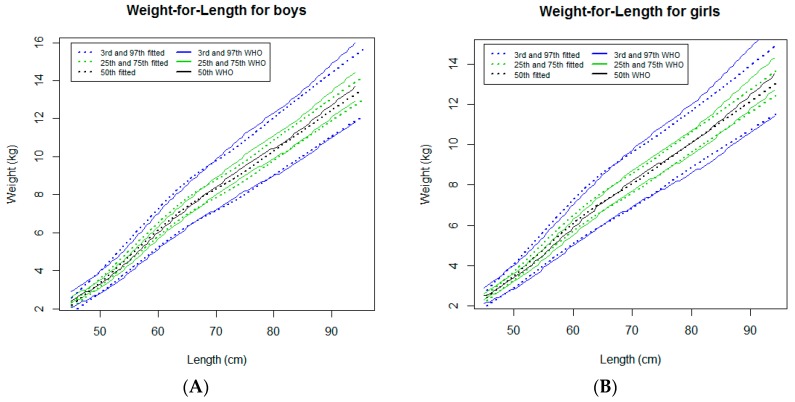
Comparison of the WHO Child Growth Standards (solid lines) with smoothed centile curves (dotted lines) of measured weight-for-length for boys (**A**) and girls (**B**) from the neonatal period to 24 months of age.

**Figure 5 ijerph-16-01693-f005:**
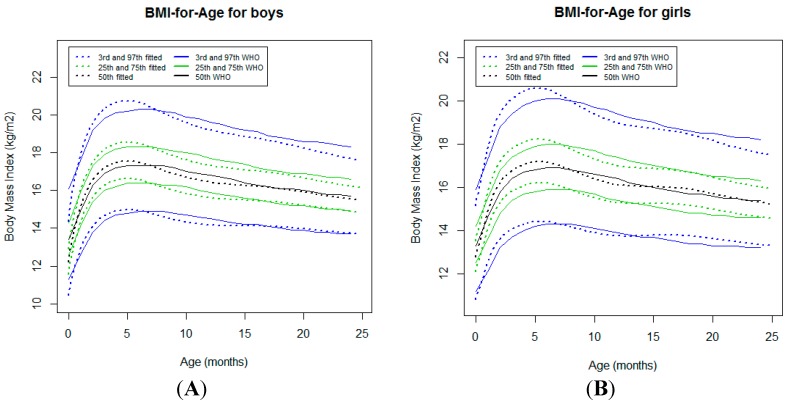
Comparison of the WHO Child Growth Standards (solid lines) with smoothed centile curves (dotted lines) of measured body mass index (BMI) for age for boys (**A**) and girls (**B**) from the neonatal period to 24 months of age.

**Figure 6 ijerph-16-01693-f006:**
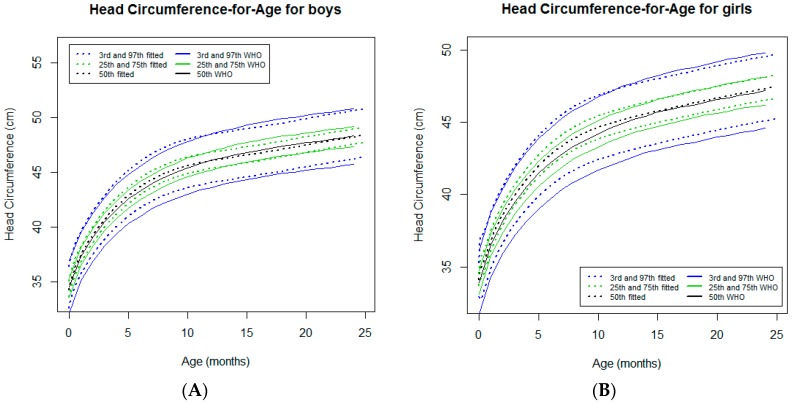
Comparison of the WHO Child Growth Standards (solid lines) with smoothed centile curves (dotted lines) of measured head circumference for age for boys (**A**) and girls (**B**) from the neonatal period to 24 months of age.

**Figure 7 ijerph-16-01693-f007:**
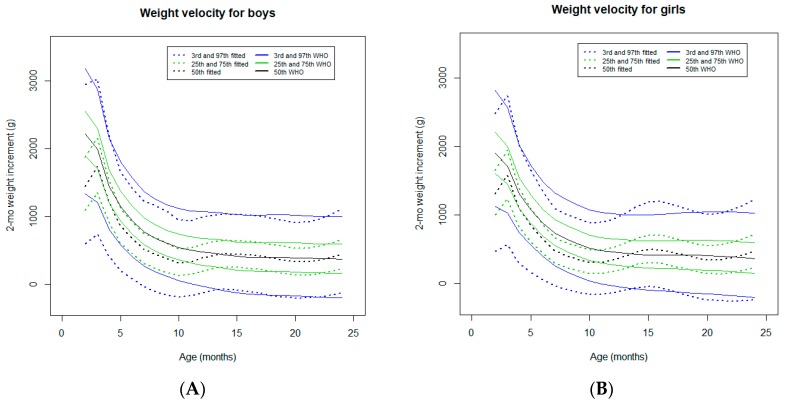
Comparison of the WHO Child Growth Standards (solid lines) with smoothed centile curves (dotted lines) of measured 2-month weight velocity for boys (**A**) and girls (**B**) from the neonatal period to 24 months of age.

**Figure 8 ijerph-16-01693-f008:**
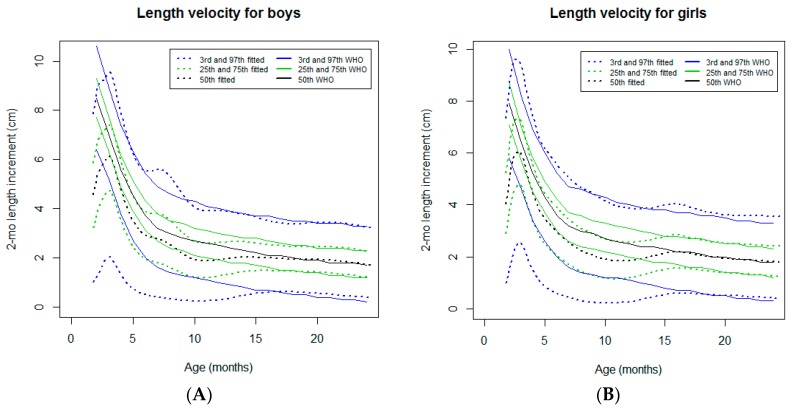
Comparison of the WHO Child Growth Standards (solid lines) with smoothed centile curves (dotted lines) of measured 2-month length velocity for boys (**A**) and girls (**B**) from the neonatal period to 24 months of age.

**Table 1 ijerph-16-01693-t001:** Differences between WHO percentiles and smoothed centile curves at 24 months of age for length-for-age, weight-for-age, weight-for-length, body mass index-for-age, and head circumference-for-age, for boys and girls

	Sex	3rd	25th	50th	75th	97th
Weight-for-age (kg)	Boys	0.1	0.5	0.7	0.8	1.2
	Girls	0.3	0.3	0.5	0.8	1.4
Length-for-age (cm)	Boys	1.8	1.8	1.7	1.7	1.7
	Girls	1.8	1.8	1.8	1.8	1.8
Weight-for-length (kg/cm)	Boys	−0.4	−0.1	0.1	0.2	0.3
	Girls	−0.4	0	0.2	0.4	0.8
BMI-for-age (kg/m^2^)	Boys	0	0	0.2	0.4	0.6
	Girls	−0.2	0	0.1	0.3	0.6
HC-for-age (cm)	Boys	−0.5	−0.3	0.1	0.3	0.2
	Girls	−0.5	−0.3	−0.1	0	0.2

HC—head circumference; BMI—body mass index.

**Table 2 ijerph-16-01693-t002:** Percentages of infants above the 90th percentile and below the 10th percentile of the WHO standards for weight-for-age, length-for-age, BMI-for-age, and head circumference-for-age, at neonatal period, 6 months, 12 months, and 24 months of age.

		At Neonatal Period		At 6 Months		At 12 Months		At 24 Months	
	Sex	Above 90th (%)	Below 10th (%)	Above 90th (%)	Below 10th (%)	Above 90th (%)	Below 10th (%)	Above 90th (%)	Below 10th (%)
Weight-for-age (kg)	Boys	14.12	10.73	9.59	8.90	4.79	22.60	3.60	14.39
	Girls	13.89	7.78	8.22	16.44	27.59	17.24	1.41	13.38
Length-for-age (cm)	Boys	17.51	4.52	2.07	14.48	4.11	21.23	3.65	18.98
	Girls	16.67	5.56	2.05	13.01	2.78	20.14	3.52	22.54
BMI-for-age (kg/m^2^)	Boys	9.04	2.82	13.79	5.52	6.16	14.38	5.84	8.03
	Girls	10.0	7.22	14.38	9.59	5.56	11.81	4.93	7.75
HC-for-age (cm)	Boys	40.68	0	18.49	2.74	5.48	7.53	5.04	4.32
	Girls	38.33	0	16.44	2.05	12.41	3.45	5.63	4.23

HC—head circumference; BMI—body mass index.

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
