# Peer review of "Comparison of Growth Curve Estimates of Infants in São Tomé Island, Africa, with the WHO Growth Standards: A Birth Cohort Study"

_ijerph, 2019, doi:10.3390/ijerph16101693_

Round 1
Reviewer 1 Report
The manuscript represents a descriptive study based on the data from the birth cohort study from São Tomé island, which is one of the African lower-middle-income countries. The authors estimated infant growth (from approximately 0.3 months to 24 months of age) using several longitudinal growth measures, and compared it to WHO growth standards.
The study is conducted in an appropriate way and methods, results and discussion are clear and supported by the data.
My specific comments are as follows:
- The authors should include the reason why 25 children were excluded from the sample of 500 eligible children.
- Were children included in this study singletons? Children from multiple pregnancies usually have different growth patterns compared with singleton births.
- Is the ethnic background of the population included in the study homogeneous? This needs to be specified and discussed accordingly.
- The part on the socioeconomic status, feeding practices and morbidities describe the population under study and should be removed to the results section.
- Why the authors decided to stop the flow chart at 5 months when the children were followed up until 24 months?
- What was the mean number and range of weight, length and head circumference measurements per child?
- Software and version used to perform the analyses should be stated in the Statistical analyses section.
- The authors should discuss the possible explanations for the decline in the weight-for age after 6 months of age in children from their cohort compared with WHO standards. Actually beyond 6 months of age all percentiles were below the WHO standards, although as authors emphasized this decline was the most pronounced for children with weight at the highest percentiles in the first 6 months of age.
- As the authors correctly state, length for age was always below the WHO standards, with similar differences from WHO standards for the whole period observed. However, from around 1 year of age (although being constant) the median of length for age for children from this cohort overlaps the 25’ percentile of WHO standards. Also, 1.8 cm difference has no the same meaning at birth and at 24 months, where I expect to see a wider range in length measurements compared to birth measurements.
- Why the authors did not estimate the weight velocity curves? Length growth velocity is the most important measure of growth in adolescence. However, weight velocity captures better the overall growth during the first years of life, as it reflects more environmental influences and conditions than the genetic potential at this early age. I suggest including also weight velocity curves.
- Not only maternal but also paternal genetic potential might have contributed to the attained length and weight of the infants, and this needs to be discussed.
- It would be interesting to report the % of children that were above 90th and below 10th percentile by the WHO standards at some pre-defined time points (for example 6 months, 12 months, 24 months). This information would better reflect growth deviations in children from São Tomé island compared with WHO standards.
- The authors speculate that the similar growth as that of the WHO standards may be due to breastfeeding practice. This could be further explored by stratifying the analyses by exclusive breastfeeding status in the first 6 months of age. Would the low numbers of not exclusively breastfed infants allow estimating growth in this subgroup of children? If yes, the authors should consider including this analysis in the manuscript.
- Similarly, it would be interesting to see whether the patterns of growth, especially weight-for age and weight velocity differ by deprivation index.
Minor comments:
- Abstract: It is not clear what does it mean „excellent weight gain“. Please reformulate this part of the sentence.
- Abstract: the sentence „The infants were born with low length z-scores and had an early postnatal velocity spurt...“ – you mean all infants of the sample had low length z-scores?
- page 15 strengths: please correct “whoch” to “which”
Author Response
The manuscript represents a descriptive study based on the data from the birth cohort study from São Tomé island, which is one of the African lower-middle-income countries. The authors estimated infant growth (from approximately 0.3 months to 24 months of age) using several longitudinal growth measures, and compared it to WHO growth standards.
The study is conducted in an appropriate way and methods, results and discussion are clear and supported by the data.
My specific comments are as follows:
- The authors should include the reason why 25 children were excluded from the sample of 500 eligible children.
Response: As specified in a previous study (Garzón 2018), 25 neonates were excluded from the birth cohort due to preterm birth, low birth weight, lack of gestational age information, major congenital malformation, and perinatal asphyxia. The reasons for exclusion are now clearly stated in the revised manuscript.
- Were children included in this study singletons? Children from multiple pregnancies usually have different growth patterns compared with singleton births.
Response: Thank you for the question and the opportunity for better explaining the criteria for inclusion. The criteria used for the construction of WHO growth standards curves (WHO 2006) were used in our cohort; thus, only singleton term appropriate-for-gestational age neonates have been included. This is now better specified in the revised manuscript.
- Is the ethnic background of the population included in the study homogeneous? This needs to be specified and discussed accordingly.
Response: Thank you for the question and the opportunity for addressing the ethnic background of the studied sample. In the revised manuscript, it is now specified that all enrolled infants were native Africans. Nevertheless, the WHO standards are recommended as a norm to monitor the growth of every child worldwide regardless ethnicity. This was already addressed as rationale in Introduction section (section 1.2.).
- The part on the socioeconomic status, feeding practices and morbidities describe the population under study and should be removed to the results section.
Response: According to the suggestion, the socioeconomic status, feeding practices and morbidities of the studied sample were moved from Methods section to the Results section in the revised manuscript.
- Why the authors decided to stop the flow chart at 5 months when the children were followed up until 24 months?
Response: Due to a typo error, the Figure 1 with the flow chart was inadvertently cut. To avoid further problems, the complete Figure 1 is included in the last page of the revised manuscript.
- What was the mean number and range of weight, length and head circumference measurements per child?
Response: In the revised manuscript, the median, minimum and maximum for weight-for-age, length-for-age, weight-for-length, body mass index-for-age, and head circumference-for-age measured in each point of assessment is now addressed in Results (section 3.2.), and detailed data included in Supplementary Tables 3a to 3e. The median was chosen instead of the mean due to the high amount of skewness of measurements’ distribution.
- Software and version used to perform the analyses should be stated in the Statistical analyses section.
Response: Software and version used to perform the analyses are now specified in Statistical Methods (section 2.5) in the revised version.
- The authors should discuss the possible explanations for the decline in the weight-for age after 6 months of age in children from their cohort compared with WHO standards. Actually beyond 6 months of age all percentiles were below the WHO standards, although as authors emphasized this decline was the most pronounced for children with weight at the highest percentiles in the first 6 months of age.
Response: Thank you for the comment. In fact, beyond 6 months of age all weight-for age percentiles became below the WHO standards and this is now correctly stated in Results and Discussion sections of the revised manuscript. The two possible explanations for decline after 6 months of age are addressed in Discussion (section 4.2.): “the suboptimal weight gain may be attributable to the possible poor quality of complementary foods (not assessed) and/or to suboptimal socioeconomic status, considering that poor wealth status is associated with decreased attained growth in children from LMICs”.
- As the authors correctly state, length for age was always below the WHO standards, with similar differences from WHO standards for the whole period observed. However, from around 1 year of age (although being constant) the median of length for age for children from this cohort overlaps the 25’ percentile of WHO standards. Also, 1.8 cm difference has no the same meaning at birth and at 24 months, where I expect to see a wider range in length measurements compared to birth measurements.
Response: Thank you for the comment. Effectively, differences from WHO standards were lower at birth than at 24 months of age and are not similar in the whole study period. It was only at 24 months of age where differences with WHO standards were constantly around 1.8 cm for all percentiles, as stated in Results section (3.2.2. of revised manuscript): “… the length-for-age curves tracked below to the WHO standards from the neonatal period to 24 months of age (Figure 3). At this age, no differences between sexes were identified for any percentiles and the differences with WHO standards were constantly around 1.8 cm for all percentiles”.
- Why the authors did not estimate the weight velocity curves? Length growth velocity is the most important measure of growth in adolescence. However, weight velocity captures better the overall growth during the first years of life, as it reflects more environmental influences and conditions than the genetic potential at this early age. I suggest including also weight velocity curves.
Response: Thank you for the suggestion. We agree that weight velocity captures better the overall growth during the first years of life. According to the suggestion, weight velocity curves were included in the revised manuscript.
- Not only maternal but also paternal genetic potential might have contributed to the attained length and weight of the infants, and this needs to be discussed.
Response: We agree that paternal height may influence offspring’s linear growth. However, this analysis is beyond the scope of the present study which was focused on the comparison of longitudinal growth of studied infants with the WHO standards. Nevertheless, it has been described that a biological interaction between the intrauterine environment and maternal inherited characteristics suppresses the influence of paternal genes (Rice 2010).
- It would be interesting to report the % of children that were above 90th and below 10th percentile by the WHO standards at some pre-defined time points (for example 6 months, 12 months, 24 months). This information would better reflect growth deviations in children from São Tomé island compared with WHO standards.
Response: As suggested, the percentage of infants above the 90th and below the10th percentiles in the WHO standards for weight-for-age, length-for-age, body mass index-for-age, and head circumference-for-age, at neonatal period, 6 months, 12 months, and 24 months are presented in Results section (Table 2 of the revised manuscript).
- The authors speculate that the similar growth as that of the WHO standards may be due to breastfeeding practice. This could be further explored by stratifying the analyses by exclusive breastfeeding status in the first 6 months of age. Would the low numbers of not exclusively breastfed infants allow estimating growth in this subgroup of children? If yes, the authors should consider including this analysis in the manuscript.
Response: In fact, only 11.6% were not exclusively breastfed at 6 months of age and the small dimension of this subgroup of individuals precludes the suggested stratification of the analyses.
- Similarly, it would be interesting to see whether the patterns of growth, especially weight-for age and weight velocity differ by deprivation index.
Response: The influence of socioeconomic deprivation on growth is beyond the scope of the present study which is focused on the comparison of longitudinal growth of studied infants with the WHO standards. This association has been analyzed in a previous study on this cohort (Garzón 2018). The multivariable analysis showed that a poor wealth status (high multidimensional poverty index - MPI) was associated with a decrease in attained growth (weight-for-length z-score, length-for-age z-score, and length-for-age difference), but not with weight velocity or length velocity z-scores (Garzón 2018).
Minor comments:
- Abstract: It is not clear what does it mean „excellent weight gain“. Please reformulate this part of the sentence.
Response: We would like to mean that in the first 6 months of age, weight gain tracked close or slightly above the WHO standards. As suggested, we reformulated this part of sentence replacing ”excellent weight gain” with “the weight gain was adequate”, meaning that weight gain was adequate in relation to the WHO standards.
- Abstract: the sentence „The infants were born with low length z-scores and had an early postnatal velocity spurt...“ – you mean all infants of the sample had low length z-scores?
Response: Thank you for the comment and the opportunity to correct the statement. In fact, the present study is not based on z-scores (as it was in the previous study by Garzón 2108). In the revised manuscript the sentence has been reformulated “Median length at birth was below median WHO standards…”.
- page 15 strengths: please correct “whoch” to “which”
Response: Thank you for the correction. The typo error has been corrected in the revised version.
References
- Garzón, M.; Pereira-da-Silva, L.; Seixas, J.; Papoila, A. L.; Alves, M. Subclinical enteric parasitic infections and growth faltering in infants in São Tomé, Africa: A birth cohort study. Int. J. Environ. Res. Public Health, 2018, 15, pii:E688.
- Rice F, Thapar A. Estimating the relative contributions of maternal genetic, paternal genetic and intrauterine factors to offspring birth weight and head circumference. Early Hum. Dev., 2010, 86, 425-432.
- WHO Multicentre Growth Reference Study Group. WHO Child Growth Standards: Length/Height-for-Age, Weight-for-Age, Weight-for-Length, Weight-for-Height and Body Mass Index-for-Age: Methods and Development. Geneva: World Health Organization, 2006.

Reviewer 2 Report
This is a very interesting paper and compares the the growth of infants in a low middle income country of about 8% of the term born children with the WHO growth standards, in the presence of two protective factors (breastfeeding and timely introduction of solids).
Some major concerns are:
1. The introduction does not immediately focuses on the main topic of the paper and therefore is somewhat confusing. In the 1st alinea, it is explained why the WHO growth standards are to be preferred above another reference, but this seems to be a methodological choice and could also be explained in the Methods section or in the second alinea. On the other hand the 2nd alinea seems to refer better to the topic as described above. I suggest to rearrange the introduction of the paper.
2. The figures show that in Sao Tome weight trajectories track close to WHO standards during the first period of life, but that a decline is shown later in life.
However, the comparison of the growth curves are only shown visually. Performing statistical tests to compare the curves with the WHO-curves during several consecutive age intervals is to be preferred to show evidence if the curves indeed differ or not.
Minor comments:
Lay-out and language should be improved on some points:
- Many pages are printed in landscape in stead of in portrait
- Some spelling errors have been made, such as 'whoch' in stead of 'which'.
- Some sentences are grammatically wrong
Ad abstract: line 19 --> what is meant by restrictive environment? Is this the womb? And if so, why is this more restrictive than the postnatal environment?
Ad figure 1 --> Half of the figure has disappeared. In addition, more details in this figure are needed, e.g. which children of the 96 missings at the 1 month visit is paying a visit at age 2 months etc.
Ad 2.2: It is not explained which visits were scheduled, where and when exactly.
Figures 2-4 and Figure 6 are difficult to read as the curves are very close to each other. Maybe editing these figures could improve this?
Ad Methods: Are data available of these 475 children on receiving breast feeding (and how long they recieved it?), when solids were introduced, and of their socio-economic status? If so, it should be considered to correct the analyses for these variables.
In the Discussion it is mentioned that downward deviations of the weight curves of girls growing in the high percentiles may be explained by the relative neglect of girls. However, wordlwide usually high weight-for-length or high BMI is not considered as a healthy development. It should be explained better why the authors consider this downward deviation as unfavorable.
Generally, the authors assume that two or three protective factors are responsible for the relatively favorable growth of the children in this country. However, other explanations for this positive result should be considered as well, such as selection bias (so: which children participated in this study?).
Author Response
This is a very interesting paper and compares the growth of infants in a low middle income country of about 8% of the term born children with the WHO growth standards, in the presence of two protective factors (breastfeeding and timely introduction of solids).
Some major concerns are:
1. The introduction does not immediately focuses on the main topic of the paper and therefore is somewhat confusing. In the 1st alinea, it is explained why the WHO growth standards are to be preferred above another reference, but this seems to be a methodological choice and could also be explained in the Methods section or in the second alinea. On the other hand the 2nd alinea seems to refer better to the topic as described above. I suggest to rearrange the introduction of the paper.
Response: Thank you for the suggestion, giving the opportunity to improve the Introduction. In the revised manuscript, the main topic is now addressed in the 1st alinea (1.1.) and the rationale for comparing with the WHO growth standards addressed in 2nd alinea (1.2.). The citations and reference list of the revised manuscript were renumbered in accordance.
2. The figures show that in Sao Tome weight trajectories track close to WHO standards during the first period of life, but that a decline is shown later in life.
Response: Thank you for the comment. In fact, beyond 6 months of age weight-for-age tracked below the WHO standards. This is now correctly stated in the revised manuscript.
However, the comparison of the growth curves are only shown visually. Performing statistical tests to compare the curves with the WHO-curves during several consecutive age intervals is to be preferred to show evidence if the curves indeed differ or not.
Response: Thank you for the suggestion. To provide a complete insight it would be necessary to analyze five percentiles in each of the 18 points of assessment. As we only have available the WHO percentile values and not raw data, the statistical analyses would originate a multiple testing problem. Due to the limited sample size of our cohort, particularly after 18 months of age, the power of the statistical approach would certainly decrease.
Minor comments:
Lay-out and language should be improved on some points:
- Many pages are printed in landscape instead of in portrait
Response: Pages printed in landscape was the only way that we found to show side-by-side charts of boys and girls for the same anthropometric measurement, thus allowing its visual comparison. A more suitable rearrangement may be edited by the Editor.
- Some spelling errors have been made, such as 'whoch' in stead of 'which'.
Response: Thank you for the correction. The typo error has been corrected in the revised version.
- Some sentences are grammatically wrong
Response: As the authors are not English native speaking, the manuscript has undergone English language editing by the MDPI (ID: english-8307). A certificate has been provided by MDPI stating that text has been checked for correct use of grammar and common technical terms and edited to a level suitable for reporting research in a scholarly journal. Anyway, English language of the revised manuscript was edited again by the MDPI (ID: english-9527).
Ad abstract: line 19 --> what is meant by restrictive environment? Is this the womb? And if so, why is this more restrictive than the postnatal environment?
Response: Thank you for the comment. In the Abstract “restrictive” meant unfavorable environment restricting fetal growth. This is reported to be mostly due to utero-placental insufficiency, but also secondary to maternal undernutrition in lower-mid-income countries (WHO 2002, Victora 2010, Christian 2013). In our cohort, we speculate that median length at birth lower than the WHO charts followed by an early length velocity spurt is explained by a transition from a restrictive prenatal environment to a more favorable postnatal environment. For not exceeding the limit of words allowed for the Abstract (200), “restrictive” was the adjective found to synthetize in one word the restrictive intrauterine milieu and underlying mechanisms affecting fetal growth. Nevertheless, to avoid confusion the term “restrictive” was replaced with “unfavorable” in the revised Abstract.
Ad figure 1 --> Half of the figure has disappeared. In addition, more details in this figure are needed, e.g. which children of the 96 missings at the 1 month visit is paying a visit at age 2 months etc.
Response: Due to a typo error, the Figure 1 with the flow chart was inadvertently cut. To avoid further problems, the complete Figure 1 is included in the last page of the revised manuscript.
Ad 2.2: It is not explained which visits were scheduled, where and when exactly.
Response: Thank you for the questions and opportunity to clarify these issues. In the revised version these details, already explained in a previous study (Garzón 2018), are now briefly specified in Methods section (section 2.1.) of the revised manuscript.
Figures 2-4 and Figure 6 are difficult to read as the curves are very close to each other. Maybe editing these figures could improve this?
Response: Thank you for the comment. According to the suggestion, all figures have been improved.
Ad Methods: Are data available of these 475 children on receiving breast feeding (and how long they received it?), when solids were introduced, and of their socio-economic status? If so, it should be considered to correct the analyses for these variables.
Response: Thank you for the question. In revised Results section (3.1.1.), data available on socioeconomic status is specified and the feeding practices are better explained.
The influence of these variables on infant growth is beyond the scope of the present study, which is focused on the comparison of longitudinal growth of studied infants with the WHO standards. Nevertheless, in a previous study on this cohort (Garzón 2018), the multivariable analysis showed that breastfeeding was positively associated with attained growth, reflected by an increase of 0.48 weight-for-length z-score and 0.39 length-for-age z-score. In multivariable analysis, no association was found between the introduction of complementary foods and infant growth (Garzón 2018).
In the Discussion it is mentioned that downward deviations of the weight curves of girls growing in the high percentiles may be explained by the relative neglect of girls. However, worldwide usually high weight-for-length or high BMI is not considered as a healthy development. It should be explained better why the authors consider this downward deviation as unfavorable.
Response: We completely agree that having a high weight-for-length or a high BMI is not considered as a healthy development. Regarding the downward deviations of the weight curves in girls growing in the high percentiles, we have stated neither that this was unfavorable nor provided any clinical judgment on this finding. Based on the literature, we simply tried to explain this finding that affected mainly girls.
Generally, the authors assume that two or three protective factors are responsible for the relatively favorable growth of the children in this country. However, other explanations for this positive result should be considered as well, such as selection bias (so: which children participated in this study?).
Response: Thank you for addressing this question and the opportunity for acknowledging this selection bias, which has been highlighted in a previous paper (Garzón 2018). In relation to infants who prematurely dropped-out, those completing the study had higher weight and length at birth, belonged predominantly to a district with higher wealth index, and their mothers were older and received longer education. In the revised manuscript, this was added in Results section and in Discussion as a limitation (section 4.5.).
References
- Christian P, Lee SE, Donahue AM, Adair LS, Arifeen SE, Ashorn P, et al. Risk of childhood undernutrition related to small-for-gestational age and preterm birth in low- and middle-income countries. Int J Epidemiol. 2013, 42, 1340-55.
- Garzón, M.; Pereira-da-Silva, L.; Seixas, J.; Papoila, A. L.; Alves, M. Subclinical enteric parasitic infections and growth faltering in infants in São Tomé, Africa: A birth cohort study. Int. J. Environ. Res. Public Health, 2018, 15, pii:E688.
- Victora CG, de Onis M, Hallal PC, Blossner M, Shrimpton R. Worldwide timing of growth faltering: revisiting implications for interventions. Pediatrics, 2010, 125, e473–e480.
- World Health Organization. Programming of chronic diseases by impairing fetal nutrition. Geneva: World Health Organization; 2002.
